# Data Mixing Optimization for Supervised Fine-Tuning of Large Language Models

Yuan Li [1]   Zhengzhong Liu [2]   Eric Xing [1 2]

## Abstract

Optimizing data mixtures for supervised fine-tuning (SFT) of large language models (LLMs) is critical for developing general-purpose models, yet this area remains underexplored. In this paper, we frame data mixing as an optimization problem and introduce a novel method designed to minimize validation loss. Our approach parametrizes the loss by modeling effective data transferred and leveraging scaling laws for fine-tuning. By experimenting with various small-scale data mixtures, we fit these parameters and derive the optimal weights. We provide both mathematical proofs and empirical results demonstrating that our algorithm achieves excellent overall and individual performance across all domains. Through controlled experiments, we show that models trained with our optimized weights perform on par with those using optimal weights determined via grid search, with per-domain loss only 0.66% higher than the best domain loss from grid search on average. Additionally, we show that reweighting popular SFT datasets using our method improves both validation loss and downstream performance. Finally, we discuss how our method can generalize to guide data selection for domain-specific models and provide insights into SFT.

## 1. Introduction

Large Language Models (LLMs) are general-purpose systems capable of performing a wide range of tasks, such as following instructions and solving mathematical problems. To train these models, a critical factor is to construct training datasets that contain diverse domains (e.g., math, code,

---

[1]Carnegie Mellon University [2]Mohamed bin Zayed University of Artificial Intelligence. Correspondence to: Yuan Li <yuanli4@andrew.cmu.edu>, Eric Xing <Eric.Xing@mbzuai.ac.ae>.

*Proceedings of the 42$^{nd}$ International Conference on Machine Learning*, Vancouver, Canada. PMLR 267, 2025. Copyright 2025 by the author(s).

healthcare)—a procedure termed *data mixing*. Data mixing has significant impacts on models' performance (Albalak et al., 2023; Xie et al., 2024). However, it is impractical to identify the best data mixture by exhaustively trying every possible combination of domain weight.

Prior research has concentrated on optimizing data mixtures during the pre-training of LLMs. A prevalent strategy for determining optimal weights of pre-training data involves defining or parametrizing the loss as a function of domain weights and subsequently minimizing this function. This overarching strategy encompasses various approaches. One such approach employs small proxy models to optimize domain weights, which are then applied to train larger models (Xie et al., 2024; Fan et al., 2024). (Xie et al., 2024; Fan et al., 2024). Alternatively, methods like Data Mixing Laws and RegMix (Ye et al., 2024; Liu et al., 2024) parameterize the relationship between model performance (e.g., loss) and data mixtures. These methods involve employing a function to estimate the performance and then using the function to optimize the data mixture.

Despite significant efforts in pre-training, data mixing for supervised fine-tuning (SFT) remains underexplored. This oversight is unwarranted, as recent research indicates that SFT not only aligns the model stylistically but also infuses knowledge and enhances the model's capabilities (Wu et al., 2024; Raghavendra et al., 2024). Moreover, existing data mixing methods designed for pre-training are not directly applicable to SFT due to several challenges: **1)** Optimizing domain weights using small proxy models may not translate effectively to larger models. Since SFT builds upon models pre-trained with distinct datasets, it is questionable whether the optimal weights derived from proxy models remain valid for the specific large-scale models at SFT stage; **2)** Even when the same model is used, parameterizing the loss without accounting for the interplay and scaling effects of different domains can lead to poor extrapolation. At the SFT stage, the goal is to train a general-purpose model with robust performance across multiple domains. Existing loss minimization does not ensure adequate representation for each domain. In fact, these loss parametrizations lead to unbalanced domain weight distributions when scaling up data sizes.

These challenges call for a systematic data mixing method for SFT. In this paper, we introduce **Data Mixing Optimization**, a method that frames data mixing as an optimization problem to determine optimal domain weights for given data budgets. Our approach involves a novel parameterization of validation loss tailored for SFT. This parameterization accounts for both the scaling effects of fine-tuning data and the interactions between different domains. We analyze our optimization method and demonstrate that our method offers two key advantages. First, it shares similar intuition and principles with existing methods (Xie et al., 2024; Fan et al., 2024; Kang et al., 2024) in up-weighting domains that effectively reduce the loss. Second, it is theoretically robust because we rigorously proved that our algorithm ensures balanced performance across all domains.

We conducted extensive experiments across various scenarios, including controlled studies using three distinct domains: instruction following, math, and code; and experiments of re-weighting popular SFT collections. Our results show that our method effectively derives domain weights that minimize validation loss. Furthermore, defining domains as specific tasks (e.g., math, code) can make our method useful for improving downstream performance on specific tasks. We also show that our data mixing optimization can be extended to guide the training of domain-specific LLMs and provide insights into SFT.

To summarize, our contributions are three-fold:

- **Data Mixing Optimization for SFT**: We introduce a method that optimizes domain weights for SFT data. Central to this optimization is a novel parameterization of validation loss that models the scaling effects of SFT data and domain interactions. This method is theoretically solid and ensures no domain will underperform.
- **Extensive Empirical Results on Data Mixing**: We present extensive SFT results about data mixing across various LLM classes and model sizes. The findings not only validate the effectiveness of our method but also reveal interesting patterns, such as the similarity in optimal weights among models within the same LLM classes, that pave the way for future research.
- **Implications and Insights for SFT**: We demonstrate that data mixing optimization can guide data selection for specialized tasks and serve as a framework to understand key aspects of SFT, including data scaling and domain interactions.

## 2. Related Work

Due to space limitations, we briefly discuss related work here and leave detailed analysis in Appendix A.

**Data Mixing.** The composition of training data for LLMs has drawn significant interest due to its impact on performance. Data mixing, also known as *domain reweighting*, is well-studied at the pre-training stage. Methods like DoReMi (Xie et al., 2024) and DoGE (Fan et al., 2024) identify optimal weights using smaller models and subsequently apply these weights to train larger models. Other approaches, such as Data Mixing Laws (Ye et al., 2024), RegMix (Liu et al., 2024), and Aioli (Chen et al., 2024a), estimate loss as a function of domain weights and interpolate performance to determine optimal data compositions. However, these methods often struggle to generalize across different model scales or data distributions, limiting their applicability to SFT. To the best of our knowledge, SMART (Renduchintala et al., 2024) is the only method explicitly designed to optimize domain weights for SFT; however, it relies on prompt embeddings and remains model and scale-invariant. In contrast, our proposed approach determines optimal domain weights that depend on both model size and scale, thus avoiding the limitations of previous methods.

**Supervised Fine-Tuning.** SFT, or instruction fine-tuning, involves training pre-trained LLMs with instruction-output pairs (Wei et al., 2022; Chung et al., 2024; Peng et al., 2023). Notable SFT collections include FLAN (Wei et al., 2022), the Tulu series (Wang et al., 2023; Ivison et al., 2023; Lambert et al., 2024), OpenHermes (Teknium, 2023), Infinity Instruct (Zhao et al., 2024), and Orca (Mukherjee et al., 2023), which aggregate data from diverse domains such as math, code, and dialogue to enhance model capabilities. According to *Superficial Alignment Hypothesis* (Zhou et al., 2024), the primary role of SFT is to align the model's formatting and style. However, recent studies (Raghavendra et al., 2024) argue that improvements from SFT follow a power law, suggesting that SFT enhances more than just style alignment, including knowledge infusion. Additionally, factors such as the number of tasks, model size, instruction settings, and output length significantly influence LLMs' performance (Puri et al., 2023; Chung et al., 2024; Longpre et al., 2023; Zhao et al.). Unlike existing methods that focus on dataset quality, our work optimizes data mixtures within specified budgets across to enhance SFT.

## 3. Data Mixing Optimization

In this section, we formulate data mixing as an optimization problem. The objective is to determine the optimal data composition that minimizes the validation loss given a fixed computational budget (i.e., training dataset size in tokens). We introduce a novel algorithm to solve this optimization problem and derive the optimal weights for data domains.

### 3.1. Problem Formulation

Consider fine-tuning an LLM using a training dataset $\mathcal{D} = \bigcup_{i=1}^{K} \mathcal{D}_i$, where $\mathcal{D}_i$ corresponds to domain $i$, and $K$ denotes the total number of domains. We similarly denote validation dataset as $\mathcal{D}^{val} = \bigcup_{i=1}^{K} \mathcal{D}_i^{val}$. Let $N = |\mathcal{D}|$ represent the total number of tokens in $\mathcal{D}$ and $N_i = |\mathcal{D}_i|$ be the number

of tokens in domain $i$. We introduce domain weight vector $\mathbf{w} = (w_1, w_2, \ldots, w_K)$, where each weight $w_i$ represents the proportion of domain $i$. The weight vector $\mathbf{w}$ satisfies the constraints of the $K$-dimensional probability simplex:

$$\mathbf{w} \in \Delta^K = \left\{ \mathbf{w} \in \mathbb{R}^K \; \middle| \; w_i \geq 0 \; \forall i, \; \sum_{i=1}^{K} w_i = 1 \right\}.$$

Each domain weight is calculated as $w_i = \frac{N_i}{N}$.

The model parameters $\boldsymbol{\theta}$ are trained by minimizing the empirical loss over the training dataset $\mathcal{D}$, yielding the parameters $\boldsymbol{\theta}^* = \arg\min_{\boldsymbol{\theta}} L(\boldsymbol{\theta}, \mathcal{D})$. We use $\boldsymbol{\theta}^*$ and $\boldsymbol{\theta}^*(N, \mathbf{w})$ interchangeably to emphasize its dependence on data composition $(N, \mathbf{w})$. After training, the model $\boldsymbol{\theta}^*$ is evaluated on the validation set $\mathcal{D}^{val}$ by computing the aggregated loss across all validation domains[1]:

$$L(\boldsymbol{\theta}^*, \mathcal{D}^{val}) = \sum_{i=1}^{K} L(\boldsymbol{\theta}^*, \mathcal{D}_i^{val}).$$

The objective is to seek the domain weight vector $\mathbf{w}$ such that, after training on the weighted domains, the model's validation loss is minimized. We formalize this into an optimization problem:

$$
\begin{aligned}
\min_{\mathbf{w}} \quad & L\big(\boldsymbol{\theta}^*(N, \mathbf{w}), \mathcal{D}^{val}\big) \\
\text{subject to} \quad & 0 \leq w_i \leq 1 \quad \forall i \in \{1, 2, \ldots, K\}, \\
& \sum_{i=1}^{K} w_i = 1, \\
\text{where} \quad & \boldsymbol{\theta}^*(N, \mathbf{w}) = \arg\min_{\boldsymbol{\theta}} L(\boldsymbol{\theta}, \mathcal{D}).
\end{aligned}
\tag{1}
$$

There is no closed-form solution for the optimization problem 1. Specifically, finding the optimal $\mathbf{w}^*$ involves an iterative optimization process. In each iteration, the model is trained using a specified weight vector, and the weights are adjusted based on the validation loss to enhance performance. However, this procedure is computationally expensive, as each iteration necessitates retraining the entire LLM. To address this issue, we propose an estimation of validation loss through the ideas of effective data from transfer (Hernandez et al., 2021) and scaling law for fine-tuning (Zhang et al., 2024).

### 3.2. Effective Data from Transfer

*Effective data from transfer* refers to the equivalent amount of training data that a model would need to achieve the same

---

[1]We compute the total validation loss across all domains, assigning equal weights to each to develop a general-purpose LLM that excels in multiple areas. Alternatively, weighting multipliers can be applied to individual domains to prioritize model performance in specific areas. We will discuss fine-tuning domain-specific LLMs in Section 5.4.

loss when trained exclusively on data from different distributions (Hernandez et al., 2021). For domain $i$, the total effective data can be calculated as the sum of in-domain data, denoted as $|\mathcal{D}_i|$, and effective data transferred to domain $i$, denoted as $|\mathcal{D}_i^T|$: $|\mathcal{D}_i^E| = |\mathcal{D}_i| + |\mathcal{D}_i^T|$.

Inspired by Hernandez et al. (2021), we represent $|\mathcal{D}_i^T|$ as a function of the dataset size from other domains $|D_{\backslash i}|$:

$$|\mathcal{D}_i^T| = k_i \cdot |D_{\backslash i}|^{\alpha_i} \tag{2}$$

where $k$ is a constant associated with the model architecture and data distribution differences, and $\alpha_i$ is a scaling coefficient. We apply Equation (2) to estimate the equivalent dataset size transferred from other domains when calculating the domain validation loss.

### 3.3. Scaling Law for Fine-Tuning

We estimate the validation loss as a function of the effective dataset size. According to Zhang et al. (2024), the scaling law for fine-tuning data follows a power-law relationship:

$$L(D) \approx C \cdot |D|^{-\beta} + E, \tag{3}$$

where $L$ is the loss, and $C$ encompasses factors such as model size. In our setting, since these factors are held constant, $C$ acts as a multiplicative term. $\beta$ is the scaling coefficient, and $E$ represents the irreducible loss. This scaling law for fine-tuning suggests that loss decreases rapidly with an initial increase in data but gradually plateaus as the dataset size grows. These implications align with previous studies that fine-tuning with a small number of examples can achieve decent results (Zhou et al., 2024), while larger fine-tuning datasets continue to enhance downstream performance (Raghavendra et al., 2024).

Leveraging effective data from transfer in Equation (2) and scaling law for fine-tuning in Equation (3), we estimate the validation loss for domain $i$ as:

$$
\begin{aligned}
\tilde{L}(\boldsymbol{\theta}^*, \mathcal{D}_i^{val}) &\approx C_i \cdot (N_i + N_i^T)^{-\beta_i} + E_i \\
&\approx C_i \cdot \big(N_i + k_i \cdot |\mathcal{D}_{\backslash i}|^{\alpha_i}\big)^{-\beta_i} + E_i
\end{aligned}
\tag{4}
$$

Here, $\mathcal{D}_{\backslash i}$ represents the aggregated datasets from all domains except domain $i$, and $|\mathcal{D}_{\backslash i}| = N - N_i$. $\beta_i$ is a scaling coefficient, and $C_i$, $k_i$ , and $E_i$ are domain-specific constants.

With this formula, we fit five parameters for each domain in Equation (4) using only small amount of data. Motivated by Kang et al. (2024), we vary the data size for domain $i$ while keeping the data sizes of other domains $|\mathcal{D}_{\backslash i}|$ constant. Specifically, the data size for domain $i$ is varied five times, denoted as $N_i^{(t)}$ for $t = 0, 1, 2, 3, 4$, where $t = 0$ corresponds to the original allocation $N_i$. For each $t$, the corresponding validation loss $L_i^{(t)}$ is recorded. To fit the

parameters, we follow the standardized procedure in Hoff-mann et al. (2024) and Zhang et al. (2024) and minimize the residuals using Huber loss (Huber, 1964) with $\delta = 0.001$. We further use Trust Region Method (Conn et al., 2000) with a nonlinear constraint (Wächter & Biegler, 2006) to ensure that the effective data from transfer $k_i \cdot (N - N_i)_i^\alpha$ remains less than the original quantity $N - N_i$. The objective is formalized as:

$$\min_{C_i, k_i, \alpha_i, \beta_i, E_i} \quad \sum_{t=0}^{4} \text{Huber}_\delta \left( \tilde{L}(N_i^{(t)}) - L_i^{(t)} \right) \quad (5)$$
$$\text{subject to} \quad k_i \cdot |N - N_i|^{\alpha_i} \le |N - N_i|,$$

where $\tilde{L}(N_i^{(t)}) = C_i \cdot \left( N_i^{(t)} + k_i \cdot |N - N_i^{(0)}|^{\alpha_i} \right)^{-\beta_i} + E_i$.

### 3.4. Optimization Algorithm

Using the fitted parameters for each domain, we formulate the optimization problem for a given data budget $N_0$ and domain weight vector $\mathbf{w}$ as follows:

$$\min_{\mathbf{w}} \quad \sum_{i=1}^{K} L\left( \boldsymbol{\theta}^*(N_0, \mathbf{w}), \mathcal{D}_i^{\text{val}} \right)$$
$$\approx \min_{\mathbf{w}} \quad \sum_{i=1}^{K} \left[ C_i \left( w_i N_0 + k_i \left( N_0 - w_i N_0 \right)^{\alpha_i} \right)^{-\beta_i} + E_i \right]$$
$$\text{subject to} \quad 0 \le w_i \le 1 \quad \forall i \in \{1, \dots, K\},$$
$$\sum_{i=1}^{K} w_i = 1$$
$$(6)$$

Note that the objective is a convex function (proof of convexity in Appendix B) but has no closed-form solution. To this end, we solve this optimization to find optimal weight vector $\mathbf{w}$ using sequential least squares programming (SLSQP) algorithm (Nocedal & Wright, 2006), a gradient-based method that handles both equality and inequality constraints. At each iteration, SLSQP constructs a quadratic approximation of the objective function $\sum_{i=1}^{K} \left[ C_i \left( w_i N + k_i (N - w_i N)^{\alpha_i} \right)^{-\beta_i} + E_i \right]$ using gradient information to refine the solution.

In summary, we integrate the aforementioned components and present the algorithm for determining the optimal data weights in Algorithm 1.

## 4. Interpretations and Analysis

In this section, we interpret our data mixing algorithm by examining key estimated parameters from perturbation experiments and analyzing mathematical implications.

**Settings.** We consider a scenario with three distinct domains: general instruction following (IF) (sampled

---

**Algorithm 1** Data Mixing Optimization

**Input:** Data budget $N_0$; sample size $N$ ($N \ll N_0$); number of domains $K$; data sources $S_1, \dots, S_K$; validation sets $\mathcal{D}^{val} = \{\mathcal{D}_1^{val}, \dots, \mathcal{D}_K^{val}\}$; perturbation ratios $r_1, \dots, r_4$.
# Initialization
**for** $i = 1$ **to** $K$ **do**
    $\mathcal{D}_i \leftarrow \text{Sample}(S_i, \frac{N}{K})$
**end for**
Train an initial model on $\mathcal{D} = \bigcup_{i=1}^{K} \mathcal{D}_i$; let $\theta^*$ be the trained model.
$\mathcal{L}_v^{(0)} \leftarrow L(\theta^*, \mathcal{D}^{val})$
# Perturbation Experiments
**for** $t = 1$ **to** 4 **do**
    **for** $j = 1$ **to** $K$ **do**
        $\mathcal{D}_j^{(t)} \leftarrow \text{Sample}(S_j, r_t \frac{N}{K})$
        $\mathcal{D}^{(t)} \leftarrow (\mathcal{D} \setminus \mathcal{D}_j) \cup \mathcal{D}_j^{(t)}$
        Train a model on $\mathcal{D}^{(t)} \rightarrow \theta^{(t)}$
        $\mathcal{L}_v^{(t)} \leftarrow L(\theta^{(t)}, \mathcal{D}^{val})$
    **end for**
**end for**
# Parameter Estimation
**for** $i = 1$ **to** $K$ **do**
    Solve for $\{C_i, k_i, \alpha_i, \beta_i, E_i\}$ via Trust Region:
$$\min_{C_i, \dots, E_i} \sum_{t=0}^{4} \text{Huber}_\delta(\tilde{L}(N_i^{(t)}) - L_i^{(t)}),$$
$$\text{subject to} \quad k_i |N - N_i|^{\alpha_i} \le |N - N_i|.$$

**end for**
# Domain Weight Optimization
Solve for $\mathbf{w} = (w_1, \dots, w_K)$ via SLSQP:
$$\min_{\mathbf{w}} \sum_{i=1}^{K} \left[ C_i \left( w_i N_0 + k_i (N - w_i N_0)^{\alpha_i} \right)^{-\beta_i} + E_i \right],$$
$$\text{subject to } 0 \le w_i \le 1, \quad \sum_{i=1}^{K} w_i = 1.$$

**Output:** $\mathbf{w}^*$

---

from Infinite-Instruct (BAAI, 2024)), math (sampled from OpenMathInstruct-2 (Toshniwal et al., 2024)), and code (sampled from OpenCoder (Huang et al., 2024)). Each domain has a unit sample size of $n = 660{,}000$ tokens. We set perturbation ratios to 1/2, 1/3, 2, and 3, and calculate the parameters by solving Equation (5). We use Llama3.2-3B as an example.

**Loss and Parameters.** To build intuition, we first analyze the validation loss for each domain. In Figure 1(a), we present a matrix comparing the token-level perplexity (PPL) of each domain as domain data decreases from $3n$ to $n$ tokens. The most notable finding is that increasing do-

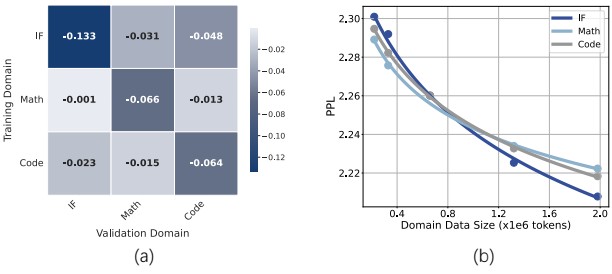

(a)                    (b)

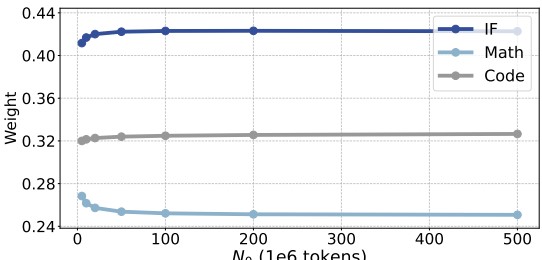

*Figure 1.* (a) Difference in per-domain perplexity between domain data sizes of $3n$ and $n$, while keeping each domain's data size fixed at $n$. (b) Overall perplexity in perturbation experiments with domain data sizes set to $1/3n$, $1/2n$, $n$, $2n$, and $3n$, while keeping each domain's data size fixed at $n$.

main data significantly reduces its validation loss. Among all domains, IF data exhibits the most substantial impact. Furthermore, increasing IF data also decreases PPL in the math and code domains. This observation aligns with our intuition during SFT: math and code problems are formatted as instructions, enhancing instruction following capabilities enables LLMs to better address these tasks.

We then explore how the loss relates to the parameters in Equation (4) using the following parametrization (repeated here for clarity):

$$\tilde{L} = C_i \cdot \left( N_i + k_i \cdot \left| N - N_i \right|^{\alpha_i} \right)^{-\beta_i} + E_i.$$

All estimated parameters for the three domains are presented in Table 1, and the fitted loss curves are shown in Figure 1(b). The parameters $\beta_i$ and $C_i$ are crucial for governing the trend of the loss curve, with $\beta_i$ serving as a decisive factor as the number of training tokens increases. We observe that $\beta_i$ for the instruction following domain data has the largest value of 0.051. This indicates that scaling up IF data is the most efficient method for reducing loss, consistent with findings in Figure 1(a), whereas math data has the minimal effect.

*Table 1.* Estimated parameters $C_i$, $k_i$, $\alpha_i$, $\beta_i$, and $E_i$ for instruction following (IF), math, and code domains.

| Domain | $C_i$ | $k_i$ | $\alpha_i$ | $\beta_i$ | $E_i$ |
|---|---|---|---|---|---|
| **IF** | 1.1562 | 0.1948 | 0.5288 | 0.0510 | 1.0967 |
| **Math** | 0.7512 | 0.0401 | 0.4467 | 0.0430 | 1.4934 |
| **Code** | 0.9820 | 0.1235 | 0.5235 | 0.0439 | 1.2679 |

**Estimated Optimal Domain Weights.** We then solve Equation (6) and plot the weights under different data budgets in Figure 2. We made two important observations: (1) *The optimal weights are scale dependent.* This finding resonates with the arguments in Kang et al. (2024) and challenges the assumption of scale independence presented in many papers. It further echoes Dong et al. (2023) by showing that different domains scale differently, resulting in relative weight adjustments based on scale; (2) *As the*

*Figure 2.* Estimated optimal weights across data budgets.

*data budget increases, the changes in weights slow down.* Domain's weight does not continue to increase or decrease significantly as the data budget increases; instead, it reaches a plateau. In other words, math data, which we discuss as being relatively less effective in reducing the loss, still maintains a decent proportion. We next analyze why this is a desirable property of our optimization algorithm to ensure that no domain will underperform.

**No Domains Left Behind.** As shown in Figure 2, estimated domain weights do not experience significant changes as the data budget increases. This is a desirable property for developing a general-purpose model that achieves strong overall performance while maintaining robust capabilities across all domains. This behavior is driven by a power-law relationship: initially, increasing data significantly reduces validation loss, but as data volume continues to scale, the marginal benefits diminish, necessitating the inclusion of data from other domains. Specifically, for each domain $i$, the partial derivative $\frac{\partial L}{\partial w_i}$ is large when $w_i$ is small and decreases as $w_i$ increases. This diminishing returns effect ensures that no single domain disproportionately dominates the training process as the data budget grows, maintaining balanced performance across all domains. We provide formal proof in Appendix C and present empirical results in Section 5 to support this argument.

## 5. Experiments

In this section, we present extensive experiments conducted under two scenarios: (1) mixing three distinct domains—IF, math, and code—given fixed data budgets; and (2) re-weighting domains for two popular SFT collections: Tulu3 (Lambert et al., 2024) and Orca (Mukherjee et al., 2023). We also evaluate models trained with re-weighted data on downstream tasks to demonstrate the practical utility of our method. Following Hernandez et al. (2021), we use token-level PPL as the primary performance metric, measured on held-out validation data for each domain. We provide experimental details, including data processing, hyperparameters, and evaluation benchmarks in Appendix D.

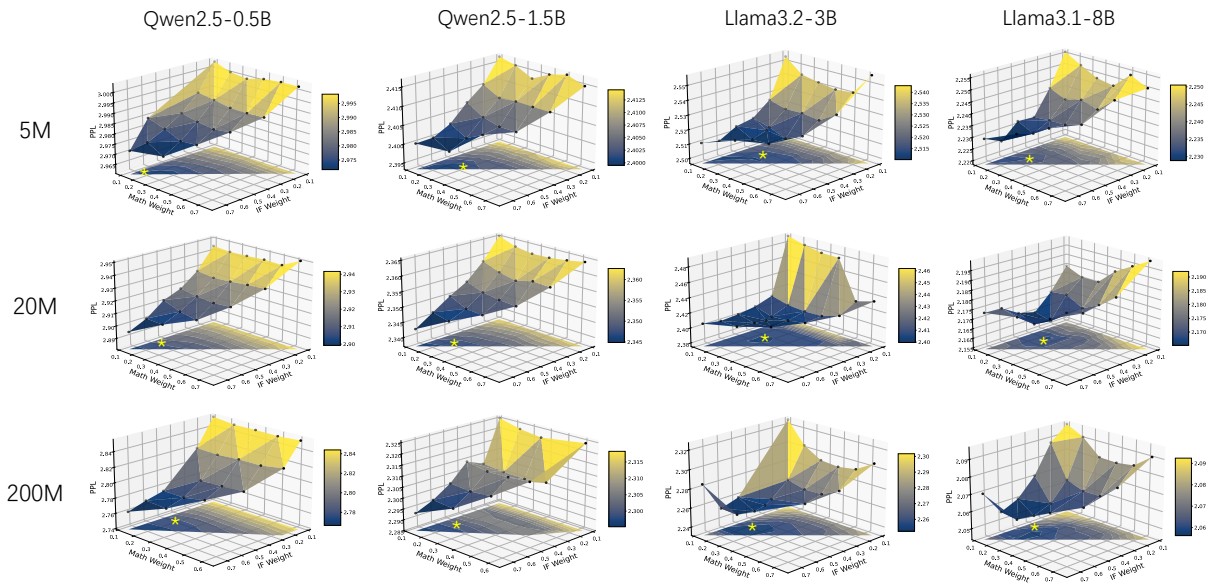

*Figure 3.* 3D surface plots of PPL for data budgets of 5M, 20M, and 200M tokens. Each plot illustrates PPL results from models trained with different domain mixture weights derived from grid search (black dots). The weights estimated by our method are highlighted with yellow stars. Only the IF and math weights are shown on the x and y axes, respectively, as the code weight is dependent.

## 5.1. Mixing Domains at Fixed Data Budgets

**Settings.** To assess our approach, we consider three distinct domains: IF, math, and code, using the same data sources as described in Section 4. For each data budget (5M, 20M, and 200M tokens), we compute the optimal weights, mix the three domains accordingly, and train the models. We use four LLMs from two classes: Qwen2.5 (0.5B and 1.5B) and Llama (3.2-3B and 3.1-8B parameters).

**Baselines.** We use data mixtures with weights determined by grid search as baselines. Specifically, we perform a grid search over the set $\{0.125, 0.25, \ldots, 0.75\}^3$, retaining only combinations where the proportions sum to 1. This results in a total of 21 valid mixtures for comparison.

**Results.** In Figure 3, we present surface plots of PPL for all 21 mixtures identified through grid search. Across different data budgets (5M, 20M, and 200M tokens) for the same model, we observe that the optimal weights are scale-dependent, i.e., the best weight configurations vary according to the size of the data budget. In addition, the weights estimated by our method (highlighted with a yellow star) effectively locate regions that achieve satisfactory, though not always optimal, performance. Training models with our weights further demonstrates that they are near-optimal. Specifically, as shown in Figure 4 (complete results in Appendix E), our model's PPL is almost on par with the best PPL by models in grid search. On average, across models and data budgets, our method's PPL is only 0.66%

higher than the best PPL. Further analysis of overall PPL and model classes is discussed in Appendix E.

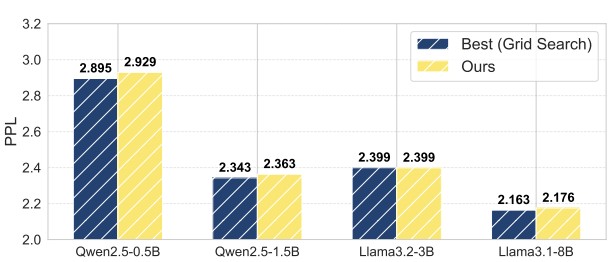

*Figure 4.* Comparison of our model's PPL with the best PPL in grid search using a data budget of 20M tokens.

We conduct a detailed investigation into PPL for each domain (all statistics in Appendix E). We compare our domain PPL with the best domain PPL across all 21 grid search mixtures. Most often, the best domain PPL is achieved when the domain's data is dominant. For Llama 3.1-8B and Llama 3.2-3B, the differences in domain PPL are not significantly greater than each other, supporting that our method can prevent any domain from underperforming. However, for the Qwen model, the IF domain shows a relatively larger deficit compared to the best IF loss in grid search than the math and code domains. This indicates that our weighting scheme, which upweights math and code relative to the best IF weights that prioritize IF, significantly affects IF domain performance. This is likely because IF is more challenging to learn for small models (as evidenced by higher domain

PPL than other domains), therefore requiring more data.

*Table 2.* Comparison of our model's domain PPL with the best domain PPL in grid search.

| Model | Scale | Domain | | |
|---|---|---|---|---|
| | | IF | Math | Code |
| | 5M | 0.054 | 0.024 | 0.005 |
| Qwen2.5-0.5B | 20M | 0.122 | 0.036 | 0.009 |
| | 200M | 0.212 | 0.043 | 0.035 |
| | 5M | 0.021 | 0.004 | -0.003 |
| Qwen2.5-1.5B | 20M | 0.061 | 0.019 | 0.020 |
| | 200M | 0.061 | 0.045 | 0.014 |
| | 5M | 0.065 | 0.049 | 0.040 |
| Llama3.2-3B | 20M | 0.053 | 0.051 | 0.035 |
| | 200M | 0.054 | 0.053 | 0.039 |
| | 5M | 0.045 | 0.036 | 0.035 |
| Llama3.1-8B | 20M | 0.039 | 0.032 | 0.054 |
| | 200M | 0.049 | 0.042 | 0.066 |

### 5.2. Re-Weighting SFT Collections

**Settings.** We further examine our method by re-weighting two popular SFT collections: Tulu3 and Orca. These collections reflect different conceptions of domains. Tulu3 identifies "skills" for LLMs and generates/collects data based on these skills. We use six of their identified skills as domains: general, knowledge recall, math, code, precise IF, and safety. Orca samples from and combines prior SFT collections, including T0, FLAN, NIV, and CoT. We follow their categorization and treat the prevalence of these datasets as domains. For domains requiring more data than originally available, we use repeated sampling. Details of data processing are shown in Appendix D.1.

**Baselines.** We compare our method with three baselines:
- **Original**: The selected instances maintain the same token proportions as the original dataset.
- **Equal_T**: All domains are weighted equally in tokens, i.e., instances sampled to ensure an equal token count from each domain.
- **Equal_I**: All domains are weighted equally with regard to the number of items.

**Results.** In Table 3, we present the domain PPL and average PPL of our method and baselines on held-out datasets. We find that our method achieves the best performance in reducing validation loss and maintains relatively low per-domain PPL across all domains, often being the lowest among all baselines. Through examining per-domain PPL, we gain insight into our method: the original Tulu3 collection assigns a dominant proportion of around 50% to math data, resulting in the lowest PPL in the math domain. However, this allocation is not the most efficient due to the diminishing returns of increasing data in a single domain,

*Table 3.* Per-domain validation loss (PPL) for reweighting Tulu3 collection.

| Weight | General | Knowledge | Math | Code | Precise IF | Safety | Avg. |
|---|---|---|---|---|---|---|---|
| *Llama3.1-8B* | | | | | | | |
| Original | 3.589 | 1.762 | 1.298 | 1.691 | 2.280 | 2.355 | 2.162 |
| Equal_T | 4.955 | 2.310 | 1.425 | 1.999 | 3.545 | 2.670 | 2.817 |
| Equal_I | 4.638 | 2.183 | 1.492 | 1.949 | 3.207 | 3.171 | 2.773 |
| Ours | 3.422 | 1.629 | 1.421 | 1.615 | 2.581 | 2.235 | **2.150** |
| *Llama3.1-70B* | | | | | | | |
| Original | 3.527 | 1.880 | 1.290 | 1.681 | 2.167 | 2.214 | 2.127 |
| Equal_T | 3.744 | 2.040 | 1.371 | 1.721 | 2.274 | 2.312 | 2.244 |
| Equal_I | 3.553 | 1.943 | 1.404 | 1.736 | 2.254 | 2.380 | 2.212 |
| Ours | 3.409 | 1.774 | 1.374 | 1.638 | 2.056 | 2.231 | **2.080** |
| *Qwen2.5-32B* | | | | | | | |
| Original | 3.701 | 1.673 | 1.262 | 1.644 | 2.276 | 2.433 | 2.165 |
| Equal_T | 3.844 | 1.734 | 1.294 | 1.645 | 2.413 | 2.546 | 2.246 |
| Equal_I | 3.891 | 1.760 | 1.268 | 1.665 | 2.390 | 2.491 | 2.244 |
| Ours | 3.891 | 1.672 | 1.292 | 1.654 | 2.047 | 2.239 | **2.133** |

as discussed in Section 4. In contrast, our method assigns a relatively smaller proportion to math and allocates more weight to domains that experience a more significant decrease in loss when scaling the same amount of data. We defer the discussion of validation loss results for the Orca collection to Appendix E, as it conveys a similar message.

### 5.3. Performance on Downstream Tasks

We have seen that our method achieves remarkable performance in decreasing validation loss. We examine whether this translate to downstream tasks with specific metrics for downstream tasks.

**Settings.** For Tulu3 collection, we also evaluation on six general tasks, each correcponds to identified skills , AGIEval (Zhong et al., 2023), IFEval (Zhou et al., 2023), MMLU (Hendrycks et al.), GSM8K (Cobbe et al., 2021), HumanEval (Chen et al., 2021), and safety (averaged over ToxiGen (Hartvigsen et al., 2022) and TruthfulQA (Lin et al., 2022)). For Orca collection, we evaluate on AGIEval (Zhong et al., 2023), HellaSwag (Zellers et al., 2019), safety (averaged over ToxiGen (Hartvigsen et al., 2022) and TruthfulQA (Lin et al., 2022)), which does not necessarily correspond to domains categorization of Orca, but it as significant overlap with the original paper (Mukherjee et al., 2023).

**Results.** We present the performance of re-weighted Tulu3 collection on downstream tasks in Table 4. We observe that our method exhibits impressive performance: for Llama3.1-8B and Llama3.1-70B, training with our weights leads to the best overall results, while for Qwen2.5-32B, it only underperforms compared to the best-performing Equal_I weight by a small margin. Additionally, we observe that our method does not cause any domain to perform significantly worse. One exception is the IFEval results for

*Table 4.* Downstream performance for different data mixing methods for re-weighting Tulu3 collection.

| Model | Weight | AGIEval | IFEval | MMLU | GSM8K | HumanEval | Safety | Average |
|---|---|---|---|---|---|---|---|---|
| Llama3.1-8B | Original | 33.9 | 54.5 | 56.4 | 65.3 | 47.0 | 43.9 | 50.1 |
| | Equal_T | 32.1 | 51.9 | 55.8 | 65.2 | 48.2 | 41.4 | 49.1 |
| | Equal_I | 32.8 | 42.7 | 54.9 | 61.9 | 50.0 | 43.9 | 47.7 |
| | Ours | 36.1 | 48.2 | 56.6 | 62.3 | 53.7 | 45.3 | **50.4** |
| Llama3.1-70B | Original | 47.7 | 65.3 | 75.4 | 86.1 | 44.5 | 47.4 | 61.1 |
| | Equal_T | 52.4 | 46.7 | 76.8 | 85.5 | 71.3 | 49.0 | 63.6 |
| | Equal_I | 51.4 | 45.8 | 76.9 | 83.9 | 51.2 | 51.8 | 60.2 |
| | Ours | 53.8 | 51.2 | 76.5 | 80.4 | 68.3 | 52.7 | **63.8** |
| Qwen2.5-32B | Original | 49.8 | 57.7 | 80.2 | 89.5 | 55.5 | 53.5 | 64.4 |
| | Equal_T | 55.8 | 61.2 | 81.0 | 89.8 | 61.6 | 56.4 | 67.6 |
| | Equal_I | 57.4 | 64.3 | 80.9 | 91.4 | 62.8 | 54.3 | **68.5** |
| | Ours | 58.3 | 61.3 | 81.0 | 90.8 | 60.5 | 56.1 | 68.0 |

Llama models, which are at least 6% lower than those with the original weights. Comparing weights of our method and original weights (full analysis in Appendix E.3), we find that that we assign more weights to domains like precise IF compared to the original. However, the total data for Tulu3 remains fixed; consequently, we have to repeatedly sample precise IF data multiple times, which might lead models to overfit to this domain. This might explains why our model performs less well on tasks like IFEval. However, we speculate that with more diverse data for the domain through synthetic data generation, our method could achieve better performance.

We show results for re-weighting the Orca collection in Appendix E.4. It provides a different story compared to the Tulu3 collection: downstream performance does not correlate with validation loss, and all methods perform similarly in downstream averages. The best-performing data weights on the held-out validation dataset do not necessarily transform to superior downstream performance here. This is not a surprise, since the evaluation can be considered out-of-distribution and is dissimilar to our held-out dataset. Additionally, because the training domains are defined by their data sources–for example, the FLAN domain internally includes various tasks and data distributions–it is difficult to determine which portion of the training data will enhance downstream performance. This emphasizes the importance of ensuring that the validation set closely matches the actual test settings. A good way to achieve this, as seen in the Tulu3 collection, is to clearly identify domains as tasks/skills, making it easier to collect similar data.

### 5.4. Extension: Training Domain-Specific LLMs.

Our formulation in Equation (6) assumes equal weighting across all domains when calculating validation loss, aligning with our objective to train general-purpose models. However, this formulation can be extended to guide data mixing for training models with one or several spe-

cialties. This is achieved by introducing domain-specific weighting factors, $\gamma_i$, transforming the objective into minimizing $\sum_{i=1}^{K} \gamma_i \cdot L(\theta^*(N_0, \mathbf{w}), \mathcal{D}_i^{\mathrm{val}})$. To illustrate this, we present a case study on training a medical chatbot, where validation dataset only consists of medical domain.

**Settings.** We mix data from two distinct domains: general instruction following (IF) and medical. General IF data are sampled from Alpaca-GPT4[2], and medical data are sampled from PubMedQA (Jin et al., 2019)[3]. The validation and testing datasets are derived from the validation and testing sets of PubMedQA. We set the data budget to 10M tokens. All other experimental settings, including unit sample size for each domain and perturbation ratios, remain consistent with previous experiments.

**Results.** Through perturbation experiments and optimization of Equation (6), we find that the optimal weights are 67.7% for Alpaca-GPT4 and 32.3% for PubMedQA. This result is somewhat unexpected, as it suggests that when the validation dataset is purely medical, our method recommends a training mixture with more general IF data than medical data to achieve optimal performance on medical tasks. This finding aligns with the *cocktail effect* (Brief et al., 2024): fine-tuning exclusively on the target task is not always the most effective strategy. Instead, inclusion of different domains—where models are trained on a mixture of related tasks—can enhance performance.

*Table 5.* Comparison of validation loss (PPL) and accuracy between our mixture and all medical data

| Metric | Our Mixture | All Medical Data |
|---|---|---|
| PPL ($\downarrow$) | 5.42 | 5.48 |
| Accuracy ($\uparrow$) | 76.8% | 76.4% |

In Table 5, we compare a mixture with weights determined

---

[2]https://huggingface.co/datasets/vicgalle/alpaca-gpt4
[3]https://huggingface.co/datasets/qiaojin/PubMedQA

by our method (67.7% IF data and 32.3% medical QA data) against a dataset of the same size composed exclusively of medical QA data. Our mixture outperforms training solely on medical data in both validation loss and downstream performance. We explain this by arguing that general IF data enhance LLMs' language understanding and response generation abilities, which are critical for answering medical questions effectively. This case study provides strong evidence that our method offers practical guidance for data mixing during the SFT stage, particularly in selecting unintuitive yet effective data combinations.

## 6. Discussion

**What is a Domain?** We follow Xie et al. (2024) and define domains based on data provenance in our experiments. For the SFT stage, this definition is nuanced: some dataset may consist of a single task, while others contain multiple tasks with diverse distributions. We do not attempt to differentiate these variations because our method, from a loss perspective, optimizes domain weights irrespective of what each domain contains—i.e., it is agnostic to the domain's composition. However, as demonstrated in Section 5.3, clearly defining domains by tasks (e.g., math, code) facilitates the selection of appropriate validation datasets for downstream tasks and streamlines follow-up data generation/collection to further enhance model capabilities.

**Future Work.** Our experiments suggest a promising direction for synergizing our method with synthetic data generation. We speculate that generating new data for domains, similar to Toshniwal et al. (2024), rather than repeatedly upsampling existing data as in our experiments, can fully reveal the potential of our approach. Another intriguing direction is to extend our parameterization of validation loss to directly model downstream performance (Isik et al., 2024; Chen et al., 2024b), thereby optimizing weights for targeted tasks.

**Conclusion.** In this paper, we introduce a novel method to optimize data mixtures for SFT of LLMs. Our method parametrizes loss as a function of domain weights while simultaneously modeling the scaling effects of fine-tuning and the interplay between domains. This design enables our method to derive optimal weights for any data scale and ensure that no domain underperforms. Extensive experimental results validate the effectiveness of our method. Furthermore, we demonstrate that this method offers valuable insights and practical guidance for SFT.

## Impact Statement

This paper presents work whose goal is to advance the field of training LLMs. There are many potential societal consequences of our work, none which we feel must be specifically highlighted here.

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

# A. Related Work

**Data Mixing.** The composition of training data in LLMs is an emerging research field because of its significant impact on performance. Most studies on data mixing, or known as *domain reweighting*, focus on the pre-training stage. Data mixtures were typically determined heuristically or through ablation studies, such as upweighting domains with higher-quality text (Touvron et al., 2023) or those distilled from advanced LLMs (Lambert et al., 2024). However, these intuitively determined weights often rely on practitioners' expertise and may not be optimal (Albalak et al., 2023). DoReMi (Xie et al., 2024) pioneered a systematic approach to data mixing by formulating it as group distributionally robust optimization, using a small proxy model to minimize excess loss and applying the resulting weights to larger models. Similarly, DoGE (Fan et al., 2024) enhances generalization by determining domain weights through optimizing gradient alignment between domains. Other methods like Data mixing laws (Ye et al., 2024), RegMix (Liu et al., 2024), and Aioli (Chen et al., 2024a) estimate the loss as a function of domain weights and interpolate performance to identify the optimal compositions.

These methods have underlying assumptions that the optimal weights are model or scale-invariant. However, weights optimized for smaller models or scales often fail to generalize to larger ones or different data distributions (Albalak et al., 2023). This issue is particularly pronounced in SFT, where models may have various pre-training data distributions. To address this, AutoScale (Kang et al., 2024) extrapolates optimal weights from small to larger scales, ensuring scale-dependent domain weighting. However, AutoScale can amplify minor weight differences, resulting in extreme or unbalanced domain weights. This may lead to the underrepresentation of certain domains, reducing generalization and effectiveness in SFT.

To the best of our knowledge, SMART (Renduchintala et al., 2024) is the only method that shares the same objective with our work to optimize domain weights for supervised fine-tuning. It uses submodular functions based on prompt embedding to determine the domain importance. However, the use of prompt embedding in SMART indicates its assumption that the optimal weights are model and scale-invariant. In contrast, our method suggests to determine optimal weights dependent on the model and scale for SFT.

**Supervised Fine-Tuning.** SFT, also known as instruction fine-tuning, refers to training pretrained LLMs with instruction-output pairs (Wei et al., 2022; Chung et al., 2024; Peng et al., 2023). SFT data is sourced from diverse domains, such as math, code, and dialogue. Wei et al. (2022) first introduced a large-scale SFT collection called FLAN, which aggregates 62 tasks in natural language inference and commonsense reasoning. Chung et al. (2024) expanded FLAN collection to over 1800+ tasks by incorporating datasets like Super-Natural Instructions (Wang et al., 2022) and PromptSource (Bach et al., 2022). Other notable open fine-tuning collections include Tulu series (Wang et al., 2023; Ivison et al., 2023; Lambert et al., 2024), OpenHermes (Teknium, 2023), Infinity Instruct (Zhao et al., 2024), and Orca Mukherjee et al. (2023), where aggregate data from various domains to enhance model capabilities.

In addition to SFT collections, current research investigates mechanisms and dataset considerations for training general-purpose LLMs. For example, Zhou et al. (2024) proposed *Superficial Alignment Hypothesis*, arguing that an LLM's capabilities are almost entirely developed during pre-training stage, and SFT is for format/style alignment. They demonstrated that carefully curated 1,000 instruction-response pairs can achieve effective instruction following abilities, with implications that the diversity and quality of the data outweigh quantity. However, Raghavendra et al. (2024) challenged this hypothesis. They contended that the improvement from SFT follows a power law; thus, SFT enhances more than formatting or style alignment, such as knowledge infusion. The importance of data diversity is widely acknowledged, supported by Puri et al. (2023) by showing an additional variant of instruction worthies many data samples. Moreover, Chung et al. (2024) and Longpre et al. (2023) analyzed factors like the number of tasks, the size of the models, and instruction settings (e.g., few-shot, chain-of-thought), and they found that these factors can have significant impact on the performance of LLMs. Zhao et al. suggested that lengthy outputs, which contain more learnable information and are harder to overfit, contribute to the success of SFT. Our work diverges by focusing not on dataset quality but on optimizing data mixtures within a given data budget and across various datasets.

# B. Proof: Convexity of Objective Function

In this section, we analyze the convexity of the objective function. Define

$$f(w) = \left( w\,N + k\,(N - w\,N)^{\alpha} \right)^{-\beta},$$

where $w \in [0, 1]$, $\alpha \in (0, 1)$, $\beta > 0$, $k > 0$, and $N > 0$. We further define

$$x(w) = w\,N\ +\ k\,(N - w\,N)^{\alpha},$$

and let

$$h(x) = x^{-\beta}.$$

Thus, our original function is $f(w) = h\big(x(w)\big) = \big[x(w)\big]^{-\beta}$. We aim to show that $f(w)$ is convex in $w$.

We first examine the concavity of $x(w)$. Observe that $x(w)$ can be rewritten as $x(s) = s + k\,(N - s)^{\alpha}$ by letting $s = w\,N$, which lies in $[0, N]$. We compute the first and second derivatives of $x(s)$ with respect to $s$:

$$x'(s)\ =\ 1\ -\ k\,\alpha\,(N - s)^{\alpha - 1},$$

$$x''(s)\ =\ k\,\alpha(\alpha - 1)\,(N - s)^{\alpha - 2}.$$

Since $\alpha \in (0, 1)$, we have $\alpha - 1 < 0$ and thus $\alpha(\alpha - 1) < 0$. Consequently,

$$x''(s)\ <\ 0,$$

indicating that $x(s)$ is concave in $s$. Because $s$ is an affine transformation of $w$, the function $x(w)$ is also concave in $w \in [0, 1]$.

Next, we check the convexity and monotonicity of $h(x) = x^{-\beta}$ for $x > 0$ and $\beta > 0$. The second derivative of $h$ is

$$h''(x)\ =\ \beta(\beta + 1)\,x^{-\beta - 2}\ >\ 0 \quad \text{for } x > 0,\ \beta > 0,$$

so $x^{-\beta}$ is convex. Furthermore,

$$h'(x)\ =\ -\beta\,x^{-\beta - 1}\ <\ 0,$$

which shows $h$ is decreasing on $(0, \infty)$.

We now apply a composition rule from convex analysis (Boyd & Vandenberghe, 2004):

**Lemma B.1** (Composition Rule for Convexity). *If $g$ is concave and $h$ is convex and non-increasing on the relevant domain, then $h \circ g$ is convex.*

In our case, $x(w)$ is concave in $w$, and $h(x)$ is convex and non-increasing in $x$. Therefore, the composition

$$f(w) = h\big(x(w)\big)$$

is convex in $w$.

To generalize, consider summing over all domains $i = 1, \dots, K$:

$$F(\mathbf{w})\ =\ \sum_{i=1}^{K} L_i(\mathbf{w})\ =\ \sum_{i=1}^{K} \Big[ C_i\,\big(w_i\,N + k_i\,(N - w_i\,N)^{\alpha_i}\big)^{-\beta_i}\ +\ E_i \Big],$$

under the constraints $w_i \geq 0$ and $\sum_{i=1}^{K} w_i = 1$. For each $\alpha_i \in (0, 1)$ and $\beta_i > 0$, then each summand $L_i(\mathbf{w})$ is convex in $w_i$. Since a sum of convex functions is also convex, the overall objective remains convex in $\mathbf{w}$. Hence one can find a global minimum of $F(\mathbf{w})$ using standard convex optimization methods. We adopt sequential least squares programming (SLSQP).

## C. Proof: No Domains Left Behind

We show that our objective function produces a balanced allocation of domain data without explicitly imposing domain-specific constraints. For convenience, recall the following optimization problem:

$$\min_{\mathbf{w}}\ \sum_{i=1}^{K} \Big[ C_i\big(w_i N + k_i(N - w_i N)^{\alpha_i}\big)^{-\beta_i} + E_i \Big]$$

$$\text{subject to}\quad 0 \leq w_i \leq 1 \quad \forall i = 1, \dots, K,$$

$$\sum_{i=1}^{K} w_i = 1.$$

We rewrite this problem in a standard constrained form as follows:

$$\min_{\mathbf{w} \in \mathbb{R}^K} \quad F(\mathbf{w}) \;=\; \sum_{i=1}^K L_i(\mathbf{w}) \;=\; \sum_{i=1}^K \Big[ C_i \big( w_i N + k_i (N - w_i N)^{\alpha_i} \big)^{-\beta_i} + E_i \Big],$$

$$\text{subject to} \quad g(\mathbf{w}) = \sum_{i=1}^K w_i - 1 = 0,$$

$$h_i(\mathbf{w}) = -w_i \le 0, \quad i = 1, \ldots, K.$$

Since $F(\mathbf{w})$ is convex (proof in Appendix B) and the set $\Delta_K = \{\mathbf{w} \in \mathbb{R}^K : \sum_{i=1}^K w_i = 1\}$ is compact, the Weierstrass extreme value theorem (Rudin et al., 1964) guarantees a unique global minimizer $\mathbf{w}^* \in \Delta_K$.

The Lagrangian is

$$\mathcal{L}(\mathbf{w}, \lambda, \{\mu_i\}) = \; F(\mathbf{w}) \; + \; \lambda \Big( \sum_{i=1}^K w_i - 1 \Big) \; - \; \sum_{i=1}^K \mu_i \, w_i,$$

where $\lambda \in \mathbb{R}$ and $\mu_i \ge 0$. By the Karush–Kuhn–Tucker (KKT) conditions, any optimal $\mathbf{w}^*$ must satisfy

$$\nabla F(\mathbf{w}^*) + \lambda^* \nabla g(\mathbf{w}^*) + \sum_{i=1}^K \mu_i^* \, \nabla h_i(\mathbf{w}^*) = \mathbf{0},$$

with complementary slackness $\mu_i^* \big( -w_i^* \big) = 0$, and $w_i^* \ge 0$, $\mu_i^* \ge 0$, $\sum_i w_i^* = 1$.

Suppose that for all $i$, we have $0 < w_i^* < 1$. Then $-w_i^*$ is strictly negative, so $\mu_i^* = 0$ by complementary slackness. Since $\nabla g(\mathbf{w}^*) = (1, \ldots, 1)$, the KKT condition becomes

$$\nabla F(\mathbf{w}^*) + \lambda^* (1, \ldots, 1) = \mathbf{0},$$

which implies

$$\frac{\partial F}{\partial w_1}(\mathbf{w}^*) \;=\; \frac{\partial F}{\partial w_2}(\mathbf{w}^*) \;=\; \cdots \;=\; \frac{\partial F}{\partial w_K}(\mathbf{w}^*).$$

Because $F(\mathbf{w}) = \sum_{i=1}^K L_i(\mathbf{w})$, each partial derivative satisfies

$$\frac{\partial F}{\partial w_i}(\mathbf{w}^*) \;=\; \sum_{j=1}^K \frac{\partial L_j}{\partial w_i}(\mathbf{w}^*).$$

In our objective function, $L_j(\mathbf{w})$ could have partial dependence on $w_i$ indirectly if $(N - w_j N)$ changes. However, each domain $i$ is governed mostly by $w_i$. That is, $\left| \frac{\partial L_j}{\partial w_i} \right| \ll \left| \frac{\partial L_i}{\partial w_i} \right|$ whenever $i \ne j$. Consequently, if some domain $i$ has a large negative gradient for small $w_i$, the optimizer increases $w_i$ until partial derivatives balance across domains.

For boundary points where some $w_i^* = 0$ or $w_i^* = 1$, complementary slackness dictates how the multipliers $\mu_i^*$ adjust the stationarity condition. If $w_i^* = 0$, then from stationarity,

$$\frac{\partial F}{\partial w_i}(\mathbf{w}^*) - \mu_i^* + \lambda^* = 0,$$

and $-w_i^* = 0$ is active, so $\mu_i^* \ge 0$. Domains $j$ with $0 < w_j^* < 1$ have $\mu_j^* = 0$, and thus

$$\frac{\partial F}{\partial w_j}(\mathbf{w}^*) + \lambda^* = 0,$$

implying

$$\frac{\partial F}{\partial w_i}(\mathbf{w}^*) = -\lambda^* + \mu_i^* \;\ge\; \frac{\partial F}{\partial w_j}(\mathbf{w}^*) = -\lambda^*.$$

Hence, if $w_i^* = 0$, domain $i$ cannot possess a strictly more negative gradient than the domains with positive allocation; otherwise, it would have been optimal to increase $w_i^*$. An analogous argument covers the case $w_i^* = 1$, showing that a domain saturates only if it truly yields the best marginal gain up to that boundary. Because each $L_i$ typically has a large negative derivative when $w_i$ is small (and a milder derivative when $w_i$ is large), the solution cannot assign zero weight to a domain that still has significantly larger marginal return than others. Similarly, in the interior case, equality of partial derivatives forces additional resources toward domains that can quickly reduce their loss, so domains with high marginal benefit do not remain under-allocated. In summary, we present the following theorem.

**Theorem C.1** (No Domains Left Behind). *The optimization problem*

$$\min_{\mathbf{w} \in \Delta_K} \sum_{i=1}^{K} L_i(\mathbf{w})$$

*admits a unique global minimizer* $\mathbf{w}^* \in \Delta_K$*, and:*

*(i) If $\mathbf{w}^*$ is in the interior of $\Delta_K$, no domain is under-allocated if it can still yield larger marginal improvement than those domains currently receiving weight.*

*(ii) If $w_i^* = 0$ for some domain $i$, then the marginal gain of allocating data to $i$ is no larger than that of any other domain $j$ with $w_j^* > 0$. Consequently, no domain with strictly higher marginal return is left at zero.*

## D. Experimental Details

### D.1. Data Processing

We provide data processing details for all datasets used in the paper below:

- Alpaca-GPT4[4]: We merged `Instruction` and `Input` as the input for the model.
- Infinity Instruct[5]: We used `0625` version. Given that the dataset encompasses both mathematical and code data, we undertook a filtering process to eliminate items featuring math-related keywords such as "algebra" and "geometry," as well as code keywords like "Python" and "Bash." This ensures the dataset remains exclusively tailored for general instruction following.
- OpenMathInstruct-2[6]: We used `train_1M` version.
- OpenCoder[7]: We used `filtered_infinity_instruct` version.
- Tulu3[8]: We filtered items larger than 4096 tokens to fit our context window. We also filter non-English data and Tulu3 hardcoded data.
- Orca[9]: We merged `System_prompt` and `Question` as the input for the model. We used 300M tokens of the original dataset.

### D.2. Hyperparamters

We use a cosine learning rate scheduler, set the batch size to 256 for training all models, and use a sequence length of 4096 tokens. For perturbation experiments, we train the model for 3 epochs and select the model with the lowest validation loss. The maximum training steps are determined by the data budget as follows: 200 steps for a 5M budget, 400 steps for 20M, and 2,500 steps for 200M. For the Tulu3 and Orca experiments, we set the maximum training steps to 6,000. All experiments are conducted on NVIDIA H100 GPUs. The learning rate for each model is show in Table 6:

### D.3. Evaluation Benchmarks

We use LM-Evluation harness[10] (with default settings if not specified) to assess models' performance on downstream tasks. Details of our downstream benchmarks are discussed below:

---

[4]https://huggingface.co/datasets/vicgalle/alpaca-gpt4
[5]https://huggingface.co/datasets/BAAI/Infinity-Instruct
[6]https://huggingface.co/datasets/nvidia/OpenMathInstruct-2
[7]https://huggingface.co/datasets/OpenCoder-LLM/opc-sft-stage1
[8]https://huggingface.co/datasets/allenai/tulu-3-sft-mixture
[9]https://huggingface.co/datasets/Open-Orca/OpenOrca
[10]https://github.com/EleutherAI/lm-evaluation-harness

*Table 6.* Learning rates for models in the experiments

| Model Class | Qwen2.5 | Qwen2.5 | Qwen2.5 | Llama3.2 | Llama3.1 | Llama3.1 |
|---|---|---|---|---|---|---|
| **Model Size** | 0.5B | 1.5B | 32B | 3B | 8B | 70B |
| **Learning Rate** | $2 \times 10^{-5}$ | $2 \times 10^{-5}$ | $2 \times 10^{-6}$ | $1 \times 10^{-5}$ | $1 \times 10^{-5}$ | $1 \times 10^{-6}$ |

- **AGIEval** (Zhong et al., 2023): AGIEval is a general benchmark. We use a specific subset of AGIEval tasks that are multiple-choice and English-only.
- **IFEVal** (Zhou et al., 2023): IFEVal measures precise instruction-following ability. We report prompt-level loose accuracy.
- **MMLU** (Hendrycks et al.): MMLU assesses knowledge across a wide range of academic subjects, including mathematics, philosophy, and law. We use a zero-shot setting.
- **GSM8K** (Cobbe et al., 2021): GSM8K evaluates mathematical problem-solving. We use an 8-shot setting with chain-of-thought prompts.
- **HumanEval** (Chen et al., 2021): HumanEval evaluates code generation capabilities. We use a zero-shot setting and report the pass@1 score.
- **ToxiGen** (Hartvigsen et al., 2022): ToxiGen assesses safety by measuring toxicity levels. We use a zero-shot setting with unnormalized accuracy.
- **TruthfulQA** (Lin et al., 2022): TruthfulQA evaluates hallucination, which we include as a measure of general safety. We use the test version of `truthfulqa_mc1` in a zero-shot setting.
- **HellaSwag** (Zellers et al., 2019): HellaSwag tests language understanding and commonsense reasoning. We report unnormalized accuracy using a zero-shot setting.

# E. Additional Results

## E.1. Further Analysis on Figure 3

In Figure 3, an intriguing observation is that models within the same class exhibit similar performance patterns concerning domain weights. We hypothesize that these models share data distributions despite differences in size. Future work could explore the relationship between pre-training and SFT data in this direction. If our hypothesis is validated, this could allow us to streamline our method by estimating loss parameters using smaller models with similar data distributions and then applying the optimized weights to larger models.

## E.2. Comparing Models Trained with Our Weights versus Grid-Search Weights

Table 7 presents the performance of models trained using our weighting method against the best results obtained through grid search. The overall PPL scores show that our method performs very close to the grid search results across all models and data budgets. This suggests that our approach is highly effective in approximating the optimal weights without the extensive computational cost of grid search. Notably, the gap between our method and grid search does not significantly expand as the data budget increases, indicating that our weighting strategy scales well with larger datasets.

Per-domain PPL results shown in Table 8 further validate the robustness of our method. Across domains including IF, Math, and Code, our weights consistently achieve performance comparable to the best grid search results. This demonstrates that our method generalizes well across diverse tasks. The consistent performance across scales and domains highlights the efficiency and reliability of our weighting approach, making it a practical guide for data mixing.

## E.3. Analysis of Weights

We present weights for each domain determined by baseline methods and our method for Tulu3 experiments and Llama3.1-70B as a study case. A key observation is the significant difference between our method and the original weights. The original weights heavily prioritize the math domain (57.77%), while drastically underweighting other domains like precise IF (2.05%) and safety (4.65%). In contrast, our method demonstrates a more balanced weighting. While it still emphasizes certain domains like safety (25.25%) and general (19.26%), it significantly reduces the weight for math (12.81%) compared to the original weights. This redistribution indicates that our method prioritizes a broader and more equitable distribution of

*Table 7.* Overall PPL of models trained with our weights and with weights in grid search.

| Model | Data Budget | Ours | Best (Grid Search) |
|---|---|---|---|
| Qwen2.5-0.5B | 5M | 2.9847 | 2.9692 |
| | 20M | 2.9293 | 2.8952 |
| | 200M | 2.8210 | 2.7621 |
| Qwen2.5-1.5B | 5M | 2.3954 | 2.3980 |
| | 20M | 2.3631 | 2.3428 |
| | 200M | 2.3154 | 2.2931 |
| Llama3.2-3B | 5M | 2.5154 | 2.5085 |
| | 20M | 2.3986 | 2.3993 |
| | 200M | 2.2536 | 2.2505 |
| Llama3.1-8B | 5M | 2.2343 | 2.2259 |
| | 20M | 2.1756 | 2.1633 |
| | 200M | 2.0762 | 2.0542 |

*Table 8.* Per-domain PPL of models trained with our weights and with weights in grid search.

| Model | Scale | IF | | Math | | Code | |
|---|---|---|---|---|---|---|---|
| | | Ours | Best | Ours | Best | Ours | Best |
| Qwen2.5-0.5B | 5M | 5.228 | 5.174 | 1.532 | 1.508 | 2.195 | 2.190 |
| | 20M | 5.120 | 4.998 | 1.506 | 1.470 | 2.162 | 2.153 |
| | 200M | 4.850 | 4.638 | 1.467 | 1.424 | 2.148 | 2.113 |
| Qwen2.5-1.5B | 5M | 3.865 | 3.844 | 1.409 | 1.405 | 1.912 | 1.915 |
| | 20M | 3.796 | 3.735 | 1.395 | 1.376 | 1.899 | 1.878 |
| | 200M | 3.661 | 3.600 | 1.393 | 1.348 | 1.892 | 1.878 |
| Llama3.2-3B | 5M | 3.734 | 3.670 | 1.630 | 1.581 | 2.182 | 2.142 |
| | 20M | 3.543 | 3.490 | 1.567 | 1.516 | 2.086 | 2.051 |
| | 200M | 3.309 | 3.255 | 1.464 | 1.410 | 1.988 | 1.949 |
| Llama3.1-8B | 5M | 3.186 | 3.141 | 1.505 | 1.469 | 2.011 | 1.977 |
| | 20M | 3.110 | 3.071 | 1.455 | 1.423 | 1.961 | 1.907 |
| | 200M | 2.919 | 2.870 | 1.402 | 1.360 | 1.907 | 1.841 |

weights across domains, in which we show in Table 4 lead to better generalization and performance across diverse tasks.

*Table 9.* Weights for each domain determined by baseline methods and our method for Tulu3 experiments using Llama3.1-70B (%)

| Method | General | Knowledge Recall | Math | Code | Precise IF | Safety |
|---|---|---|---|---|---|---|
| Original | 16.42 | 5.77 | 57.77 | 13.33 | 2.05 | 4.65 |
| Equal_T | 16.67 | 16.67 | 16.67 | 16.67 | 16.67 | 16.67 |
| Equal_I | 32.99 | 12.57 | 24.30 | 13.65 | 10.43 | 6.04 |
| Ours | 19.26 | 16.92 | 12.81 | 10.10 | 15.63 | 25.25 |

### E.4. Validation Loss and Downstream Performance of Re-Weighting Orca Collection

Table 10 and Table 11 highlight an important discrepancy between validation loss PPL and downstream task performance. In Table 10, our method achieves competitive or even superior PPL scores compared to other weighting strategies, particularly for larger models like Llama3.1-70B and Qwen2.5-32B. However, Table 11 reveals that lower PPL does not consistently translate to better performance on downstream tasks such as AGIEval, HellaSwag, and Safety. For instance, while our method achieves the lowest average PPL for Llama3.1-70B and Qwen2.5-32B, its downstream performance is not always the best, as seen with Llama3.1-8B, where the Equal_T weighting outperforms ours on average.

This inconsistency can be attributed to the distribution difference between the validation dataset and the testing dataset for downstream tasks. The validation dataset may not fully capture the complexity or diversity of real-world tasks, leading to selection of the model that does not generalize well to downstream applications. Therefore, it indicates that our method, while effective in reducing validation loss, requires careful curation of validation dataset that aligns with downstream tasks to translate its powerfulness to downstream applications.

*Table 10.* Validation loss for reweighting Orca collection.

| Weight | T0 | CoT | Flan | Niv | Avg. |
|---|---|---|---|---|---|
| *Llama3.1-8B* | | | | | |
| Equal_T | 1.782 | 1.808 | 1.948 | 1.645 | 1.796 |
| Equal_I | 1.728 | 1.813 | 1.930 | 1.580 | 1.763 |
| Original | 1.726 | 1.845 | 1.924 | 1.665 | **1.790** |
| Ours | 1.781 | 1.826 | 1.949 | 1.626 | 1.795 |
| *Llama3.1-70B* | | | | | |
| Equal_T | 1.629 | 1.679 | 1.785 | 1.571 | 1.666 |
| Equal_I | 1.608 | 1.682 | 1.770 | 1.515 | 1.644 |
| Original | 1.600 | 1.720 | 1.772 | 1.581 | 1.668 |
| Ours | 1.620 | 1.651 | 1.767 | 1.534 | **1.643** |
| *Qwen2.5-32B* | | | | | |
| Equal_T | 1.643 | 1.670 | 1.798 | 1.553 | 1.666 |
| Equal_I | 1.626 | 1.673 | 1.787 | 1.511 | 1.649 |
| Original | 1.617 | 1.699 | 1.789 | 1.560 | 1.666 |
| Ours | 1.613 | 1.662 | 1.787 | 1.533 | **1.649** |

*Table 11.* Downstream performance for reweighting Orca collection.

| Weight | AGIEval | HellaSwag | Safety | Avg. |
|---|---|---|---|---|
| *Llama3.1-8B* | | | | |
| Equal_T | 38.3 | 59.5 | 39.7 | **45.9** |
| Equal_I | 39.6 | 59.2 | 38.9 | 45.9 |
| Original | 39.7 | 59.5 | 40.3 | 46.5 |
| Ours | 38.2 | 59.9 | 39.7 | 45.9 |
| *Llama3.1-70B* | | | | |
| Equal_T | 52.6 | 67.6 | 49.4 | 56.5 |
| Equal_I | 53.7 | 67.8 | 41.9 | **54.5** |
| Original | 53.3 | 67.2 | 51.1 | 57.2 |
| Ours | 52.6 | 67.6 | 44.3 | 54.8 |
| *Qwen2.5-32B* | | | | |
| Equal_T | 59.8 | 64.4 | 58.5 | 60.9 |
| Equal_I | 61.2 | 64.3 | 59.9 | 61.8 |
| Original | 60.6 | 64.4 | 55.9 | 60.3 |
| Ours | 61.7 | 64.4 | 54.6 | **60.3** |

