# OpenReview forum: "Data Mixing Optimization for Supervised Fine-Tuning of Large Language Models"
_ICML.cc/2025/Conference — ICML 2025 poster_

### Official Review · Reviewer_wneA · 2025-03-05

**Overall Recommendation:** 5

**Summary:**

This paper introduces a way to decide the optimal proportion of different domain datasets for LLM fine-tuning. The Scaling law is utilized to learn the optimal proportions, improving fine-tuning performance in various settings.

**Claims And Evidence:**

Yes.

**Essential References Not Discussed:**

N/A

**Experimental Designs Or Analyses:**

Yes.

**Methods And Evaluation Criteria:**

Yes.

**Other Comments Or Suggestions:**

N/A - see Other Strengths And Weaknesses

**Other Strengths And Weaknesses:**

## Strengths
- The paper is very well organized and easy to read.
- The idea of using Scaling Laws to predict validation loss and therefore the optimal dataset proportions is very interesting.
- Theoretical proof is provided.
- Experiments are done thoroughly on various settings with different models and datasets.

## Weaknesses and Questions
- I did not spot any critical weakness in this manuscript. One minor concern is the relatively marginal improvement of "Ours" compared to "Original". But considering that it is hard to expect groundbreaking improvements just by changing the dataset proportion, the results are acceptable.
- Out of curiosity, will this method consider synergy effects of different domains? For instance, some arbitrary pair of domains can have synergic effects, which may impact the optimal proportion that should be assigned. Relevant analyses or discussions might be very interesting.

**Questions For Authors:**

N/A - see Other Strengths And Weaknesses

**Relation To Broader Scientific Literature:**

This work relates to data mixing literature in general. It proposes a new method by using the scaling laws to predict the optimal proportions.

**Theoretical Claims:**

Yes.

---

> ### Author Rebuttal · Authors · 2025-03-31
>
> Thank you so much for your feedback! We will discuss your comments as follows.
>
> ---
> Q:  Relatively marginal improvement of "Ours" compared to "Original"
>
> A: We have noticed this issue and, over the past month, we conducted additional experiments to understand it. Our responses are as follows: In our paper, we mentioned using repetitive sampling for domains with higher weights than the original dataset, which we speculated to be the cause of the observed downperformance, not the problem with our method. Taking the domain weights of Llama 3 70B in Tulu3 as an example (shown in Table 0.1.3 https://anonymous.4open.science/r/Data_Mixture-5D78/src/Table0.1.3.png), the "Precise IF" domain in the original Tulu3 dataset constitutes about 2%, but in our calculation, we aimed to increase it to approximately 15%. This means that, for the same amount of data, the same item in the "Precise IF" domain appears around 7x more frequently than in the original dataset when using repetitive sampling. We hypothesized that this repetitive sampling strategy might cause unintended model behaviors, potentially explaining the suboptimal performance. To address this, we replaced repetitive sampling with a re-writing method, inspired by [1] and [2]. Re-writing can be intuitively understood as paraphrasing repetitive items: it does not add new information but reduces repetition in the sampled dataset. We then trained the model on the resulting datasets and attached the downstream performance in https://anonymous.4open.science/r/Data_Mixture-5D78/src/downstream.png. From the results, we observed that our method led to more significant improvements in downstream benchmarks, validating the effectiveness of our approach.
>
> To further validate our method, we present the PPL-step plot in https://anonymous.4open.science/r/Data_Mixture-5D78/src/PPL_step.png (we use Llama 3.1-8B model as an example, with the original dataset’s weight for comparison). Our method achieves comparable performance with faster computation. Ultimately, our method converged at a lower PPL.
>
> ---
> Q: Will this method consider synergy effects of different domains?
>
> A: This is a very intriguing and insightful question. To give a brief answer: our method partially accounts for synergy effects. Specifically, it does not consider pairwise influence between domains but instead looks at the aggregated effect of all other domains on the current domain, as detailed in Section 3.2. That said, we’d be happy to share some of our experiences. Early in the project, we attempted to model pairwise synergy effects between domains, initially focusing on just two domains. Appendix F5 illustrates this with instruction following and healthcare domains. We found that instruction following data indeed enhanced performance on healthcare tasks more than healthcare data alone. This finding is interesting, and seems intuitive: effectively responding to healthcare questions requires a model to understand and follow prompts—skills that are improved by instruction following data. However, when we introduced a third domain, the intercorrelations between the two original domains changed, almost becoming unrelated. This led us to conclude that the behaviors of SFT are complex and case-specific. Nevertheless, understanding the underlying mechanisms of SFT remains both an intriguing and challenging task.

---

### Official Review · Reviewer_jCLn · 2025-03-13

**Overall Recommendation:** 3

**Summary:**

This paper addresses the problem of data mixing for supervised fine-tuning (SFT) of large language models (LLMs). The authors frame data mixing as an optimization problem and introduce a method to minimize validation loss. Their approach involves parameterizing the loss by modeling effective data transferred and leveraging scaling laws for fine-tuning. The authors provide mathematical proofs and empirical results to demonstrate the effectiveness of their algorithm.

**Claims And Evidence:**

The claims made in this submission are generally well-supported by the evidence provided. The authors present both theoretical analyses and experimental results to validate their method.

**Essential References Not Discussed:**

The paper adequately discusses essential references. However, it could benefit from a more in-depth discussion of recent work on scaling laws and their implications for LLM training.

**Experimental Designs Or Analyses:**

I have checked the soundness/validity of the experimental designs and analyses, and they seem to be well-designed and the analyses are appropriate.

**Methods And Evaluation Criteria:**

The proposed method, Data Mixing Optimization, and the evaluation criteria, including experiments on controlled data mixtures and popular SFT collections, are appropriate for the problem.

**Other Comments Or Suggestions:**

* Consider adding a section discussing the computational cost and scalability of the proposed method.

* It would be helpful to include a more detailed comparison with other data mixing techniques, highlighting the advantages and disadvantages.

**Other Strengths And Weaknesses:**

Strengths:

* The paper introduces a novel and theoretically sound method for data mixing in SFT.

* It provides a comprehensive set of experiments and results across various scenarios.

* The paper is well-written and easy to follow.

Weaknesses:

* The paper could benefit from a more detailed discussion of the limitations of the proposed method.

* The generalization of the method to different data domains (excluding IF， Math and Code) could be explored further.

**Questions For Authors:**

* Could you elaborate on how the proposed method can be extended to guide data selection for domain-specific LLMs, as mentioned in the paper? How would the parameterization of validation loss be adapted in such cases?

* In the experiments, how sensitive is the performance of the proposed method to the choice of validation dataset? Could you discuss any potential strategies for selecting an appropriate validation dataset to ensure good generalization performance?

**Relation To Broader Scientific Literature:**

The key contributions of this paper are related to the broader scientific literature through its focus on optimizing data mixtures for LLM training. It builds upon existing work in data mixing for pre-training and addresses the underexplored area of data mixing for SFT.

**Theoretical Claims:**

I have checked the correctness of the proofs and the theoretical claims appear to be sound.

---

> ### Author Rebuttal · Authors · 2025-03-31
>
> Thank you! We will address your comments as follows.
>
> ---
> Q: The generalization of the method to different data domains (excluding IF, Math and Code) could be explored further.
>
> A: In Section 5.2, we conducted experiments beyond the three domains (IF, Math, and Code), exploring additional domains such as Safety, Knowledge, and General. We also experimented domains that are not well-defined in the Orca dataset. Furthermore, in Section 5.4 and Appendix F5, we included experiments in the Healthcare domain to further validate the generalizability of our method.
>
> ---
> Q: The paper could benefit from a more detailed discussion of the limitations of the proposed method.
>
> A: Thank you for your suggestion! One limitation of our work is that our method parameterizes validation loss. Since we are in the post-training stage and targeting downstream applications, modeling validation loss is less straightforward than directly modeling downstream performance. However, modeling downstream performance is challenging due to the lack of a clear mapping between performance metrics and data size, like what’s in scaling law that connects data size with validation loss. Another limitation is that our method relies on parameter estimation, which can be sensitive at times. A potential solution is to train the model multiple times and compute the mean of validation loss when fitting parameters, though this comes with the tradeoff of increased computational cost.
>
> ---
> Q: Computation cost & scalability
>
> See Q2 and A2 under Reviewer AAeY about computation cost.
>
> Our method demonstrates strong scalability across various data sizes. In Section 5.1, we conducted experiments with 5M, 20M, and 200M tokens. In Section 5.2, the data size was approximately 500M tokens, equivalent to around 1M items. For context, industrial-level models like Qwen 2.5 and DeepSeek V3 typically use 1M–3M items for SFT. Thus, the data sizes in our experiments are quite substantial.
>
> ---
> Q: It would be helpful to include a more detailed comparison with other data mixing techniques, highlighting the advantages and disadvantages.
>
> A: While most existing data mixing techniques are applied during the pre-training stage, to our knowledge, SMART is the only method designed for the post-training stage, and we have already incorporated it in our paper. However, SMART has its limitations, as it relies on embeddings of data items and is agnostic to the models used. This is problematic because model performance is significantly influenced by the choice of base models. Other post-training data mixing efforts are either not publicly disclosed or require extensive manual effort.
>
>
> ---
> Q: How the proposed method can be extended to guide data selection for domain-specific LLMs? How would the parameterization of validation loss be adapted in such cases?
>
> A: The procedure for using our method to guide data selection for domain-specific LLMs follows the same approach as in general cases. The only difference is that the validation loss calculation is focused solely on the validation dataset for the specific domain, excluding other domain datasets. As it turns out, domains that contribute little to the validation loss for a specific domain will have very low domain weights.
>
> ---
> Q: How sensitive is the performance of the proposed method to the choice of validation dataset? Could you discuss any potential strategies for selecting an appropriate validation dataset to ensure good generalization performance?
>
> A: Good questions! In general, when there is a significant distribution difference between the validation and downstream test sets, the best performance on the validation set (typically used to select the model with lowest validation loss) is unlikely to translate to best downstream performance. This applies to our method as well. Developing effective strategies for selecting an appropriate validation dataset to ensure strong generalization is similar to those for training datasets. Key principles include broad domain coverage, diversity in prompts/templates/questions, and a balanced distribution of difficulty. For a comprehensive list of guidelines, we refer to the Kimi K1.5 technical report [4], which provides extensive criteria for data collection and filtering. Additionally, to enhance the validation dataset, synthetic data generation using LLMs can be employed to create in-distribution data. A good example is OpenMathInstruct-v2 [5], which generates high-quality, in-distribution math instruction data based on existing datasets. Similar approach can be applied to other domains as well.
>
> [4] Kimi k1. 5: Scaling reinforcement learning with llms
>
> [5] Openmathinstruct-2: Accelerating ai for math with massive open-source instruction data

---

### Official Review · Reviewer_AAeY · 2025-03-13

**Overall Recommendation:** 3

**Summary:**

This paper proposes a data mixing optimization problem for supervised fine-tuning of LLMs. The problem is based on a scaling law determined by neural scaling laws for SFT. The paper proves convexity properties about the optimization problem and applies it to optimize mixtures for several finetuning settings.

**Claims And Evidence:**

The paper claims to develop a data mixing optimization framework and empirically validate it. It is well-supported.

**Essential References Not Discussed:**

N/A

**Experimental Designs Or Analyses:**

The experiments appear standard.

**Methods And Evaluation Criteria:**

The methods and evaluation criteria make sense, and are well tested on a variety of benchmarks.

**Other Comments Or Suggestions:**

Some grammatical errors in Section 3.1 (e.g., Consider "fine-tuning")

**Other Strengths And Weaknesses:**

Strengths:
1. The paper is generally easy-to-read with key results flowing intuitively.

Weaknesses:
1. Improvements in Table 3-4, appear marginal. It is hard to discern the advantage of the proposed method in this case and whether the improvement is potentially random. Typically, we may see a plot of the validation loss over steps to show that the method allows for achieving the same performance at faster computation. Similarly, improvements in Figure 4 show that grid searching is marginally better than the proposed method, at roughly the same level of improvement as the comparison in Table 3-4. Ultimately, it is hard to discern what constitutes a meaningful performance improvement for this SFT data mixing task.

2. Similarly, it would be useful to measure overall compute cost of this approach, from estimating the scaling laws all the way to optimization. Is there a compute advantage to the proposed method? For example, since grid searching over a handful of mixing ratios seems to improve over the proposed method, but surely the proposed method is computationally better than grid searching?

**Questions For Authors:**

See Strengths + Weaknesses

**Relation To Broader Scientific Literature:**

The paper contributes to the data mixing literature by focusing on data mixing for SFT. It introduces a new method in this space.

**Theoretical Claims:**

I checked the proofs in the Appendix. They follow standard procedures and appear correct.

---

> ### Author Rebuttal · Authors · 2025-03-31
>
> Thank you! We will address your comments as follows.
>
> ---
> Q1 : Improvements appear marginal
>
> A1 : We have noticed this issue and, over the past month, we conducted additional experiments to understand it. Our responses are as follows: In our paper, we mentioned using repetitive sampling for domains with higher weights than the original dataset, which we speculated to be the cause of the observed downperformance, not the problem with our method. Taking the domain weights of Llama 3 70B in Tulu3 as an example (shown in Table 0.1.3 https://anonymous.4open.science/r/Data_Mixture-5D78/src/Table0.1.3.png), the "Precise IF" domain in the original Tulu3 dataset constitutes about 2%, but in our calculation, we aimed to increase it to approximately 15%. This means that, for the same amount of data, the same item in the "Precise IF" domain appears around 7x more frequently than in the original dataset when using repetitive sampling. We hypothesized that this repetitive sampling strategy might cause unintended model behaviors, potentially explaining the suboptimal performance. To address this, we replaced repetitive sampling with a re-writing method, inspired by [1] and [2]. Re-writing can be intuitively understood as paraphrasing repetitive items: it does not add new information but reduces repetition in the sampled dataset. We then trained the model on the resulting datasets and attached the downstream performance in https://anonymous.4open.science/r/Data_Mixture-5D78/src/downstream.png. From the results, we observed that our method led to more significant improvements in downstream benchmarks, validating the effectiveness of our approach.
>
> To further validate our method, we present the PPL-step plot in https://anonymous.4open.science/r/Data_Mixture-5D78/src/PPL_step.png as you suggested (we use Llama 3.1-8B model as an example, with the original dataset’s weight for comparison). Our method achieves comparable performance with faster computation. Ultimately, our method converged at a lower PPL.
>
> Finally, we want to emphasize that the baselines we are comparing against are already highly optimized (e.g., the best configuration from grid search in Section 5.1, and the original weights in Tulu3, which required extensive manual labor and multiple iterations to fine-tune [3]). Therefore, the advantage of our approach lies not only in performance improvement but also in its full autonomy and relatively lower computational cost, as we will explain further below.
>
> ---
> Q2: Computation cost
>
> A2: The computation cost of our approach primarily lies in running on domain samples over N tokens to determine the parameters in Eq6. We compare the computation cost of our method, and cost of running the entire training dataset in (we use deepspeed.profiling.flops_profiler to estimate overall number of floating-point operations (FLOP) as a metric).
>
> For experiments in Section 5.1 (see https://anonymous.4open.science/r/Data_Mixture-5D78/src/cost_5.1.png), our method costs computation of ~7x compared to training on 5M tokens, ~3x compared to training on 20M tokens, and 0.7x compared to training on 200M tokens. While for grid search, the current granularity runs each experiment for 21 times, which is substantially more.
>
> For re-weighting Tulu3 experiments (see https://anonymous.4open.science/r/Data_Mixture-5D78/src/cost_tulu3.png), our method costs ~1.5x compared to one training round, which is impressive.
>
> Similar stories ring true in re-weighting Orca experiments (see https://anonymous.4open.science/r/Data_Mixture-5D78/src/cost_orca.png).
>
> Therefore, our method demonstrates impressive advantages in terms of computation costs compared to traditional manual methods seen in [3], which takes extensive runs to determine the data mixture.
>
>
> [1] Wizardlm: Empowering large language models to follow complex instructions.
>
> [2] Datagen: Unified synthetic dataset generation via large language models.
>
> [3] Tulu 3: Pushing frontiers in open language model post-training.

---

### Official Review · Reviewer_rbjG · 2025-03-14

**Overall Recommendation:** 2

**Summary:**

This paper proposes an algorithm to optimize proportions of data sources for the supervised fine-tuning (SFT) stage in LLM training by minimizing validation loss. Given that jointly optimizing the mixing weights and model parameters is expensive, the paper makes an approximation. It makes uses of data scaling laws (Zhang et al. 2024) to parameterize the validation loss for each domain and borrows the ideas of 'data transfer' (Hernandez et al 2021) to come up with a 5-parameter regression model to estimate validation loss from each domain. Given this model, the mixture optimization becomes a convex function, which is solved using gradient methods.  The paper shows that the model takes into account interactions between domains and thus assigns an equitable proportion to each domain. Results using two LLM families (Qwen 2.5 and Llama 3.1) show that the method obtains slightly worse performance compared to an exhaustive grid search. The paper also shows that the method can be used to rebalance dataset mixtures in existing SFT datasets.

## update after rebuttal
In my initial review, I asked the authors to justify why they needed to perform a 2 stage regression for a 5 parameter model and added pointers to several papers (from statistics literature) which argued why that should not be done. The authors did not answer this question adequately. They mentioned that the single stage regression is necessary only when there are a large number of parameters - this is not the case: A single stage model should always yield the best performance while a 2 stage model will likely yield biased results (e.g. overinflated regression coefficients, overfitting to the training set etc). If the authors want to show that the two stage model is equivalent to a one stage model, they should report results using both models. Without such an empirical justification, I find it hard to trust the results from the paper. Hence, I am not changing my scores after the rebuttal.

**Claims And Evidence:**

Some of the claims can be improved:
* The paper should report how well the validation loss estimates works by reporting mean squared error.

**Essential References Not Discussed:**

None

**Experimental Designs Or Analyses:**

Yes, they seem reasonable.

**Methods And Evaluation Criteria:**

Some of the methods in the paper need to be improved:
* Appendix B (P656): For estimating the domain specific validation loss in Equation (4), the paper employs a two-step linear regression in whereby the first stage estimates \beta and the second step estimates the remaining variables with \beta fixed. Such a multi stage approach is referred to in statistics as 'stepwise selection' and it is known to suffer from a number of issues e.g. biases in regression coefficients, problems in the presence of multi-collinearity. See references below:

References:
* https://link.springer.com/book/10.1007/978-3-319-19425-7
* https://hbiostat.org/rmsc/multivar
* https://journalofbigdata.springeropen.com/articles/10.1186/s40537-018-0143-6

**Other Comments Or Suggestions:**

* What is the difference between 'data budget' N0 and 'sample size' N?
* In Figure 3, the PPL of the proposed method (yellow star) is always better than the grid search though this is not the case in the tables presented in Appendix F.

**Other Strengths And Weaknesses:**

Strengths:
* Proposes a method to optimize mixture ratios for SFT
* Method shows competitive but slightly weaker ratios compared to grid search
* Method can be used to rebalance mixing ratios in existing SFT datasets
* The proposed approach yields equitable mixture ratios such that no single dataset dominates the overall mixture

Weaknesses:
* It would be good to see some results on how well the validation loss approximation using the 5-parameter models holds by reporting MSE.
* Though the method yields the mixture set with a low validation set loss, this does not always translate to performance improvements in downstream evaluations.
* Some of the aspects of the method are not explained or justified clearly e.g. why is the 5-parameter regression done in 2 stages i.e. finding \beta in the 1st stage and the rest of the parameters in the 2nd stage? There is evidence from the multiple regression literature showing that a stepwise selection suffers from multiple issues (See above).

**Questions For Authors:**

None

**Relation To Broader Scientific Literature:**

The paper introduces a new technique to optimize weighting of data sources in SFT. Past work has mostly focussed on mixture optimization in the context of pre-training. The only prior work which has studied this problem for SFT is SMART (Renduchintala et al., 2024) - their approach differs from this work in using prompt embedding and the use of submodular functions.

**Theoretical Claims:**

The proofs seem reasonable.

---

> ### Author Rebuttal · Authors · 2025-03-31
>
> Thank you! We will address your comments as follows.
>
> ---
> Q:  Validation loss approximation in MSE
>
> A: The MSE of the validation loss estimation in the experiments from Section 5.1 is presented in Table 0.1.1 (https://anonymous.4open.science/r/Data_Mixture-5D78/src/Table0.1.1.png). Additionally, the MSE of our estimation for the re-weighting experiments with Tulu3 and Orca are shown in Table 0.1.2 in the link). While not perfect, our method provides a reasonable estimate of the validation loss.
>
> ---
> Q: Low validation losses do not always translate to downstream evaluations.
>
> A: In the re-weighting Orca experiments, we concluded that a low validation loss does not necessarily translate to improved downstream evaluations if the validation datasets are not properly sampled. This is intuitive and applies to many machine learning scenarios, highlighting the importance of careful validation dataset selection. Our key claim is that our method, which optimizes for validation loss, will generally lead to improved downstream performance when validation dataset is properly chosen.
>
> So now I think the primary concern lies in why the improvement in downstream performance for re-weighting Tulu3 appears marginal (in Table 4). We have noticed this issue and, over the past month, we conducted additional experiments to investigate it. Our responses are as follows: In our paper, we mentioned using repetitive sampling for domains with higher weights than the original dataset, which we speculated to be the cause of the observed downperformance. Taking the domain weights of Llama 3 70B in Tulu3 as an example (shown in Table 0.1.3 in the link)), the "Precise IF" domain in the original Tulu3 dataset constitutes about 2%, but in our calculation, we aimed to increase it to approximately 15%. This means that, for the same amount of data, the same item in the "Precise IF" domain appears around 7x more frequently than in the original dataset when using repetitive sampling if we keep the training dataset size fixed. We hypothesized that this repetitive sampling strategy might cause unintended model behaviors, potentially explaining the suboptimal performance. To address this, we replaced repetitive sampling with a re-writing method, inspired by [1] and [2]. Re-writing can be intuitively understood as paraphrasing repetitive items: it does not add new information but reduces repetition in the sampled dataset. We then trained the model on the resulting datasets and attached the downstream performance in "downstream.png" in the link. From the results, we observed that our method led to more significant improvements in downstream benchmarks, validating the effectiveness of our approach.
>
> ---
> Q: why is the 5-parameter regression done in 2 stages
>
> A: Our decision to use a two-stage estimation was driven by empirical and intuitive considerations rather than strict theoretical requirements. We have conducted experiments initially estimating all five parameters in a single stage using the Huber loss and found that the resulting domain weights are very similar to those obtained via a two-stage approach. So in our context, the impact of using one versus two stages might not affect the domain weights significantly if both estimations form similar curves of validation loss.
> We went through the materials you provided, and we noted that the issues associated with two-stage estimation are most pronounced in models with a large number of variables and when dependencies among explanatory variables are poorly captured. In our method, however, we only have 5 parameters (this might explains why multi-step estimation is not that problematic) and parameters are not treated equally. In Eq. (6), \beta exerts a much stronger influence on the validation loss when the dataset is large. Since the validation loss (from the trained model) can be noisy, we want to use additional information—specifically, the pairwise slopes across different training runs—to determine \beta more reliably.
>
> ---
> Q：'Data budget' N0 and 'sample size' N?
>
> A: The data budget N0 refers to the total number of tokens allocated for the final training dataset, which we use to optimize the distribution of domains. The sample size N is the number of tokens from each domain used to fit the parameters in Equation 4.
>
> ---
> Q: In Figure 3, the PPL of the proposed method (yellow star) is always better than the grid search though this is not the case in the tables presented in Appendix F.
>
> A: The yellow star in Figure 3 represents the points which our method identifies, and are near-optimal rather than always outperforming the grid search. The yellow stars are located in regions that correspond to good PPL values, and they are not necessarily the best PPL achieved by the grid search. Many of the yellow stars are positioned at the boundaries, slightly off the darkest areas, indicating that while our method performs well, it may not always match the best results from the grid search.

---

### Decision · Program_Chairs · 2025-05-01

**Decision:**

Accept (poster)

**Comment:**

This paper presents an approach to data mixing optimization for the supervised fine-tuning of LLMs. The authors frame data mixing as an optimization problem aimed at minimizing validation loss, introducing a method that parametrizes this loss by modeling effective data transferred and leveraging scaling laws for fine-tuning. The proposed algorithm achieves performance often on par with computationally expensive grid search methods. Furthermore, the paper showcases the practical utility of their approach by demonstrating improvements in both validation loss and downstream performance when reweighting popular SFT datasets.

During the review process, several constructive critiques were raised that the authors should address in the camera-ready version to further strengthen the manuscript. A primary concern revolved around the two-stage parameter estimation approach detailed in Appendix B. Reviewers requested a more thorough justification for this method compared to a single-stage approach and suggested providing empirical evidence of their equivalence or the benefits of the two-stage method, potentially by reporting results from both. Additionally, it was suggested to report the mean squared error (MSE) of the validation loss estimation to quantify the accuracy of the approximation. Finally, the authors should consider elaborating on the limitations of their method, including its reliance on validation loss as a proxy for downstream performance and the potential sensitivity of parameter estimation.